# High Variability and Dual Strategy in the Wintering Red Kites (*Milvus milvus*)

Jorge García-Macía [1,*], Javier De La Puente [2], Ana Bermejo-Bermejo [2], Rainer Raab [3] and Vicente Urios [1,*]

1    Vertebrates Zoology Research Group, University of Alicante, Apdo. 99, E-03080 Alicante, Spain
2    SEO/BirdLife, Bird Monitoring Unit, C/Melquiades Biencinto 34, E-28053 Madrid, Spain; jdelapuente@seo.org (J.D.L.P.); abermejo@seo.org (A.B.-B.)
3    Technisches Büro für Biologie Mag. Dr. Rainer Raab, Quadenstrasse 13, 2232 Deutsch-Wagram, Austria; rainer.raab@tbraab.at
*    Correspondence: jorgegarciamacia97@gmail.com (J.G.-M.); vicenteurios@yahoo.es (V.U.)

**Abstract:** To develop effective conservation strategies for migratory birds, it is essential to understand the ecology of a species at each biological phase, including its wintering (or non-breeding) season. For the red kite (*Milvus milvus*), an endangered raptor from the Western Palearctic, its wintering ecology is little known. We tagged 44 red kites using GPS/satellite transmitters to study their non-breeding seasons in Spain. Two spatial strategies were recorded: 34 individuals (77%) spent all their wintering periods in only one area, whilst the remaining individuals (23%) moved between two main areas at least once. This strategy, however, was not consistent over the years. In the latter case, the distance between wintering areas was $311.6 \pm 134.7$ km, and individuals usually spent equally long periods in each area ($96 \pm 35$ days). No effects of age or sex were found on these area shifts, so they may have been driven by food or habitat resource availability. We also found high interindividual variability in home range sizes. The home ranges of adults were two- to three-times smaller than those of immatures, probably due to a better knowledge of the territory.

**Keywords:** raptors; non-breeding; GPS telemetry; movement ecology; spatial ecology



## 1. Introduction

The knowledge of complete life cycles is necessary to understand the dynamics of populations and to develop effective conservation strategies [1–3]. Studies applied to raptor conservation should be focused not only on migration or breeding, but also on their wintering strategies and the wandering movements of individuals.

Satellite/GPS telemetry has increased the understanding of how individuals interact with their environment [4]. This technology has been used to study the wintering areas on many migratory raptors, such as the pallid harrier [5], the Eleonora's falcon [6,7], the Egyptian vulture [8] and the booted eagle [9]. However, there are still several gaps in the knowledge of the spatial ecology of many raptors, such as the red kite (*Milvus milvus*). The ecology of the red kite during its wintering period has been mainly studied based on field observations [10–14] and just a few studies have used GPS tracking technology [15,16].

The red kite (*Milvus milvus*, Accipitridae) is a medium-sized, partially migratory raptor from the Western Palearctic. It is a facultative colonial bird that tends to group in roosts during winter, with an average of 70 individuals per roost [17]. Kites usually hunt small animals, such as voles and other rodents, rabbits, etc. in heterogenous landscapes [18,19], but they also have scavenging habits. They frequent garbage dumps, slaughterhouses or areas where they can find carrion and other organic remains [19,20]. These scavenging habits, together with their semi-colonial behaviour, are a serious problem for the conservation of the species as a consequence of the poisons that can be found in these food sources [21].

Europe holds at least 90% of the global red kite population, with 24,000–31,900 pairs breeding in 17 countries and a winter population size of 38,100–42,200 individuals [22].

The wintering population trend is considered to be decreasing on both the short and long terms [23]. Currently, the species is listed as "Least Concern" by the International Union for Conservation of Nature and Natural Resources [24] but "Endangered" according to the Spanish List of Threatened Species. The red kite has a great variability in its spatial strategies. Most red kites in north-eastern Europe are intracontinental migrants and they spend the wintering season mainly in southern France, Spain and Portugal [25,26], but there is also a growing number of sedentary individuals in Central Europe [27], and a small number of red kites wintering in northern Africa (Eurokite, unpublished data). In addition, there are non-migratory breeding populations in the wintering areas of birds coming from the north, as is the case in Spain. Birds usually start the autumn migration to their wintering areas in southern Europe between August and November and return to their breeding areas between February and April [26,28]. According to the last census, Spain is the main winter destination for the species, with an estimated wintering population of 50,297 individuals [17]. However, the number of populations wintering outside of Spain are generally increasing [24]. Therefore, the red kite is probably changing its distribution in Europe. In some countries, breeding and wintering populations are increasing and becoming progressively more sedentary, but elsewhere, especially at the species' southern distribution limits, including Spain, which currently supports the main wintering populations, numbers are decreasing. Global warming could displace red kites to a more northern distribution, as has been happening in the last years with several species [29–31]. For this reason, the study of the variability in the wintering strategies of the red kites is mandatory in understanding the behaviour of the species, and its conservation needs elaborating accurate management plans.

We have tagged 44 red kites that breed in Central Europe and winter in Spain and studied their wintering seasons. Our main objectives are: (1) to analyse the wintering season of the red kite in Spain using parameters such as duration, beginning and end dates; (2) to locate and characterize wintering areas using minimum convex polygon (MCP) and kernel density estimators (95% KDE, 75% KDE and 50% KDE); (3) to study ranging behaviour during the winter period and check for possible different strategies according to age and sex.

## 2. Methods

### 2.1. Tagging and Data Collection

From 2013 to 2020, 44 red kites were captured during their wintering seasons in Spain, in different provinces: Álava (18), Huesca (15), Segovia (10) and Toledo (1). There were 24 adults and 20 immatures (at the time of tagging); in terms of sex, 9 were males (adults), 4 females (adults) and 31 were individuals whose sex was not determined (11 adults and 20 immatures; Table S1). We considered individuals as immature during their first wintering season, and adult from their second [26]. Birds were trapped using an automatic clap net on the ground, triggered remotely and baited with carrion. Individuals were weighed and ringed. We obtained blood samples for sex determination by DNA from 13 individuals. Birds were tagged using a GPS or satellite transmitter that was affixed to the back using a Teflon harness (a non-abrasive material) tied with a cotton thread [32,33]. The weight of the tags represented between 1.64% and 3.41% of the birds' body masses (mean = 2.28, SD = 0.43), which is below the recommended standards [34]. Birds were released a maximum of 30 min after capture.

We used four transmitter models: 20–21-g SAKER GPS-GSM (Ecotone Telemetry; $n = 17$), 22-g PTT-100 solar-powered Argos/GPS (Microwave Telemetry Inc.; $n = 2$), 18-g solar PTT 45"-bat (North Star Science and Technology; $n = 2$), 20–25-g OrniTrack-20 and OrniTrack-25 solar powered GPS-GSM trackers (Ornitela; $n = 24$). One individual (Alava 05) was tagged with two different transmitter models, as it was originally tagged with an Ecotone tag, but, after three years it was recaptured and tagged with an Ornitela one.

Ecotone tags were programmed to collect locations every hour, from 06:00 to 19:00 (local time); microwave tags collected one location every 2 h, from 06:00 to 18:00 h, except

between 1 February–31 March and 1 October–30 November with 1 loc/h; 18-g tags, with a duty cycle of 12-h ON/12-h OFF, and Ornitela tags recorded data every 5 min from nautical dawn in the morning to nautical dusk in the evening, when the battery level was sufficient. The locations were filtered at a 1-h frequency in order to homogenize the data provided by the different transmitters

### 2.2. Spatial Parameters and Analysis

For each wintering event (or wintering season), the arrival date was calculated as the day when the bird arrived at its wintering site from autumn migration, and the end date as the last day remaining at the wintering ground. We considered full-wintering events (32) but also partial ones (40), when individuals were tagged during wintering season or there were temporary transmitter failures (Table S1). In the case of immature individuals, the first wintering season was always partial, as they were captured at the wintering areas after the first autumn migration from their natal nests.

We also identified individuals that performed area shifts during their wintering seasons, spending weeks or months in a new area before spring migration to Central Europe. Both partial and complete wintering seasons were included, given that we did not detect behavioural changes in the individuals after trapping. Each of these areas were considered as a sample unit in the analyses, due to there having been no significative differences in their sizes compared with one-area winters (see Section 3).

We calculated site fidelity as the percentage of individuals that repeated their wintering areas over consecutive years, when data was available.

We estimated the home range areas for each wintering area with the minimum convex polygon (MCP) and 95%, 75% and 50% kernel density estimation (KDE). Individuals with two differences in home-range parameters were evaluated according to number of wintering areas (one, two), sex (male/female) and age (immature/adult). For this purpose, we performed 12 different linear mixed models (LMMs) including, as dependent variables, the different measurements of home range areas (MCP, 95% KDE, 75% KDE and 50% KDE). "Number of wintering areas" (one, two), "age" (immature, adult) and "sex" (male, female) were considered as fixed factors, while "individual" identity and "wintering area" as random factors. We used 82 wintering areas for modelling age and number of wintering areas, and 33 for age models.

All statistical analyses were performed with R 4.0.5 [35] and significance level was established at $p < 0.05$. Home range size estimations and LMM's were performed using the statistical packages *sf* and *lme4* for R, respectively. Maps were drawn with QGIS 3.16.6.

### 3. Results

Of the 44 tagged red kites, data were recorded for 72 wintering events (52 for adults and 20 for immatures). All individuals performed short-term intracontinental migrations, from their breeding area in Central Europe to their wintering areas in Spain. The wintering areas were located in the Spanish provinces of Huesca, Álava, Segovia, Toledo, Burgos and Cáceres, and in southern France (Figure 1, Table S1). After the spring migration towards the northeast, the countries of destination, where the red kites had their breeding areas (or where they spent the summer, in the case of immatures), were mainly Germany (44 wintering events) and Switzerland (12), but also Poland (8), France (4), Austria (1) and Belgium (1). Information about the last breeding destinations of two individuals was not available because of death or transmitter failures during migration.

The red kites spent between 32 and 139 days in their wintering areas, averaging $96 \pm 27$ days. Adults began their wintering seasons between mid-October and mid-January, on average, on 17 November (immature beginning dates were not available). Adults finished their wintering seasons between the end of January and the first week of June, on average, on 3 March, while immatures stopped wintering between the end of February and the first week of May, on average, on 6 April (Table 1).

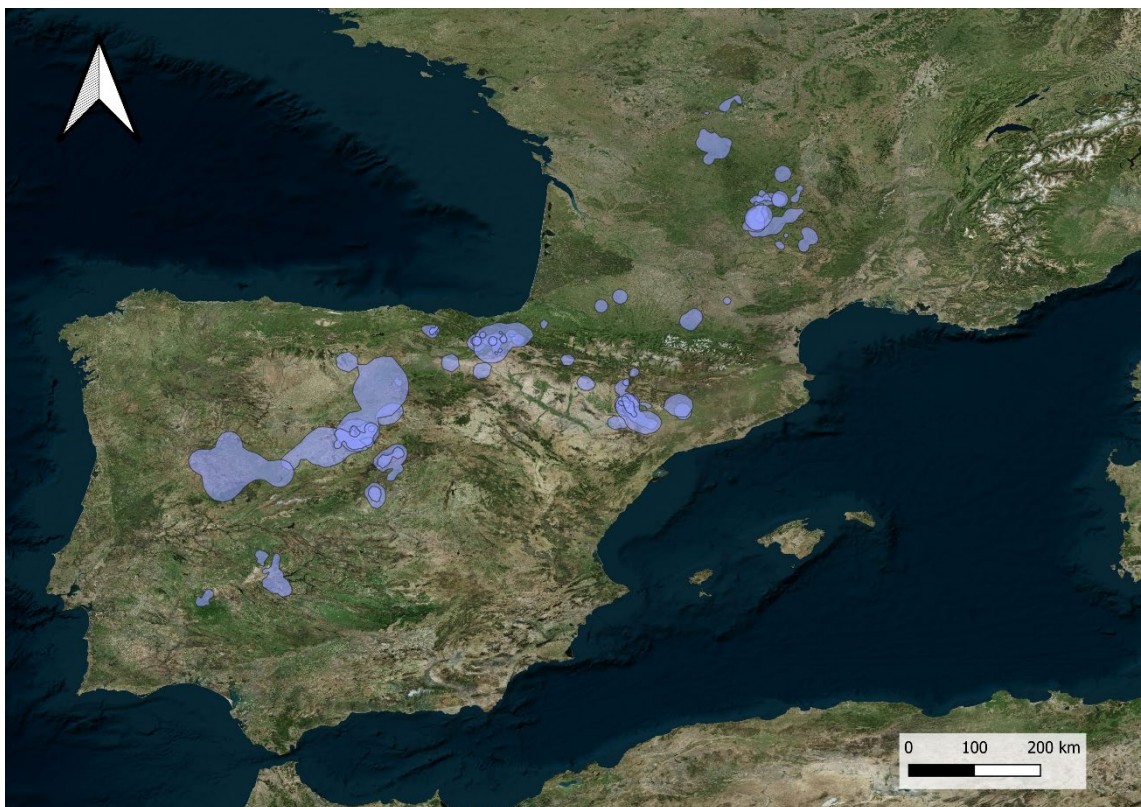

**Figure 1.** Wintering areas (95% KDE) used by the 44 tagged red kites during their non-breeding seasons in Spain and Southern France.

**Table 1.** Home range size (km$^2$) and duration of 72 wintering seasons of 44 red kites tracked by GPS satellite telemetry in Spain according to their number of wintering areas, sex, and age (adults or immatures). Results are expressed as mean and standard deviation. Minimum and maximum values appear in parenthesis. Red kites with two wintering areas were excluded from the age (adults, immatures) and sex (males, females) pools. MCP—minimum convex polygon. * Only considering complete winter, not partial.

| | *n* | Days of Wintering * | MCP (km$^2$) | 95% Kernel (km$^2$) | 75% Kernel (km$^2$) | 50% Kernel (km$^2$) |
|---|---|---|---|---|---|---|
| **overall** | 82 | 95 ± 30 (32–139) | 2173 ± 3519 (27–20340) | 1158 ± 2182 (11–13176) | 438 ± 899 (5–4904) | 158 ± 344 (0.1–2056) |
| **one area** | 62 | 96 ± 27 (33–138) | 2315 ± 3776 (27–20340) | 979 ± 2030 (11–13176) | 373 ± 810 (5–4216) | 130 ± 291 (0.1–1517) |
| **two areas (considered separate)** | 10*2 | 97 ± 34 (39–139) | 1655 ± 2377 (54–9998) | 1774 ± 2609 (45–10891) | 659 ± 1155 (11–4904) | 257 ± 482 (3–2056) |
| **adults** | 60 | 94 ± 29 (32–138) | 1614 ± 2825 (27–16827) | 796 ± 1380 (21–8550) | 271 ± 504 (5–3107) | 111 ± 230 (0.08–1518) |
| **immatures** | 22 | – – | 3586 ± 4632 (521–20340) | 2111 ± 3379 (11–13175) | 877 ± 1440 (8–4904) | 282 ± 528 (3–2056) |
| **males** | 19 | 119 ± 13 (101–127) | 2828 ± 4510 (32–16828) | 918 ± 1270 (21–4995) | 292 ± 475 (7–1959) | 98 ± 181 (0.1–743) |
| **females** | 14 | 92 ± 29 (33–139) | 830 ± 1204 (27–4381) | 646 ± 921 (53–2753) | 223 ± 323 (26–1128) | 86 ± 121 (7–396) |

These red kites had great site fidelity during non-breeding seasons: 21 out of 24 individuals with available data (89%) returned to the same wintering area over consecutive years. (Table S1).

### 3.1. Dual Strategy: One or Two Wintering Areas

We found two spatial strategies during the wintering season. Thirty-four out of 44 individuals (77%) spent all their wintering periods in only one area, whilst the remaining individuals (23%) moved between two main areas (Figure 2, Table 2). This strategy, however, was not consistent over the years, and these latter individuals always used only one area in the rest of their recorded years (when they were available). These area shifts were carried out between Spain and France, by crossing the Pyrenees ($n = 5$), between different areas within the Iberian Peninsula ($n = 4$), or within Southern France ($n = 1$). Both adults (6) and immatures (4) used two wintering areas. Individuals with one wintering area spent $96 \pm 27$ days (mean $\pm$ SD) to complete their wintering season, while individuals with two wintering areas spent $96 \pm 35$ (considering the sum of the two areas). The individuals that performed area shifts showed similar phenology to the individuals with one wintering area, beginning their wintering seasons on 24 November (range = October–January) and finishing them on 11 March (range = January–May).

The ten individuals that changed areas during winter did so mainly at the beginning of winter, after autumn migration; and at the end of winter, prior to spring migration, when they returned to their nests or summering locations in Central Europe. The distance between wintering areas was $311.6 \pm 134.7$ km (range = 108–530). Individuals usually spent equally long periods in each wintering area, without lengthening the total duration of wintering compared with other individuals or other years of the same individual (Table 2).

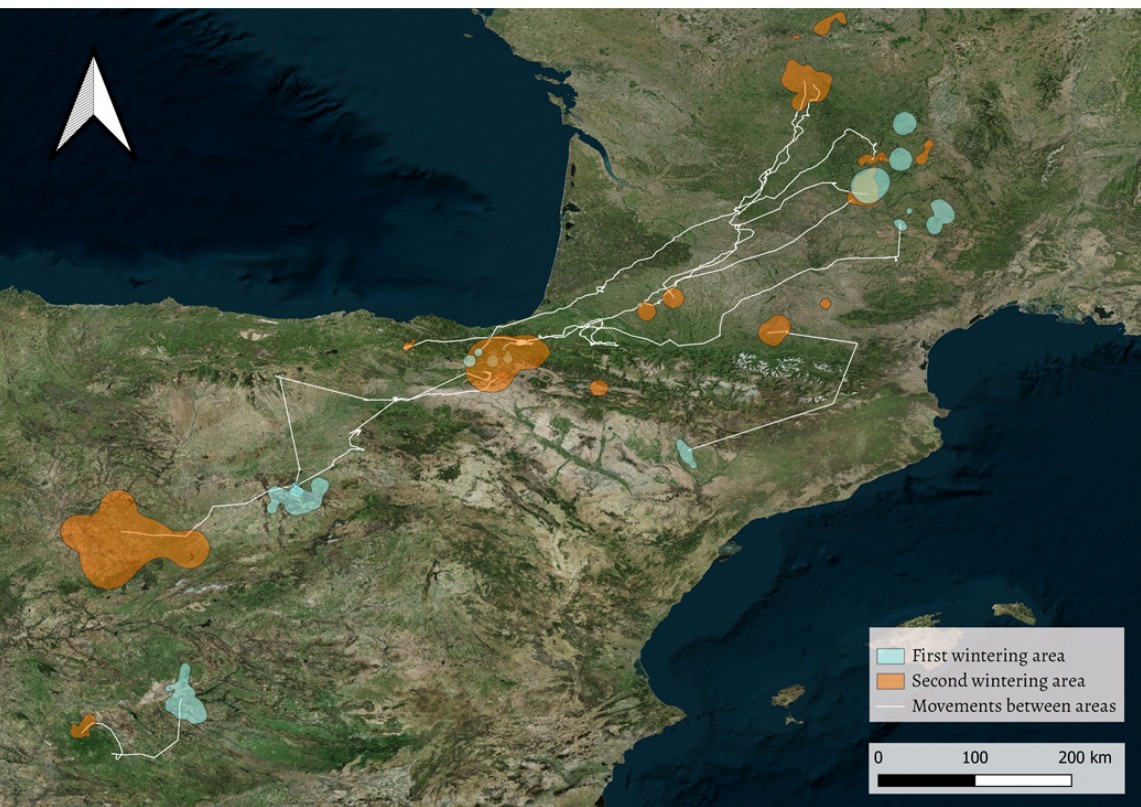

**Figure 2.** *Cont.*

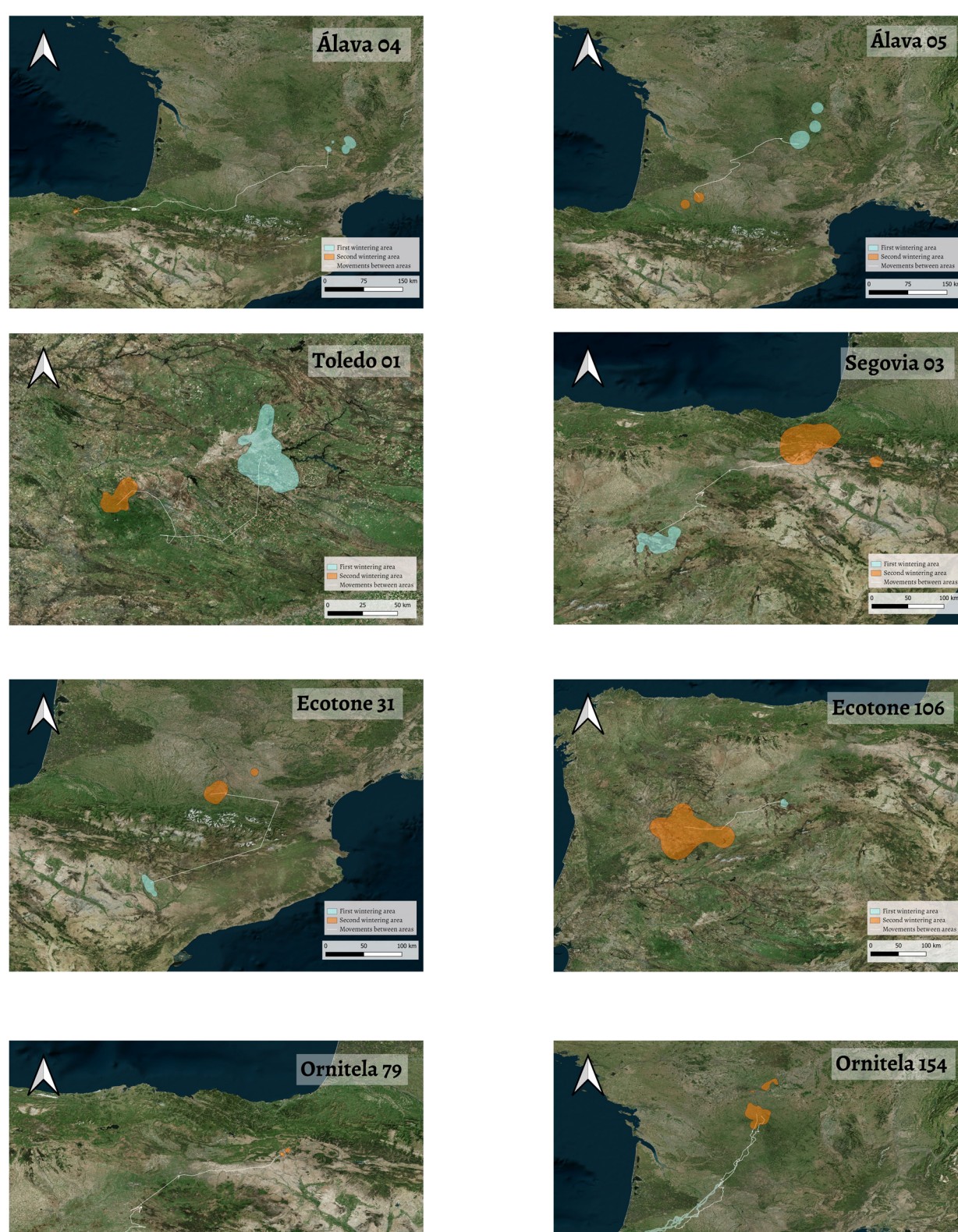

**Figure 2.** *Cont.*

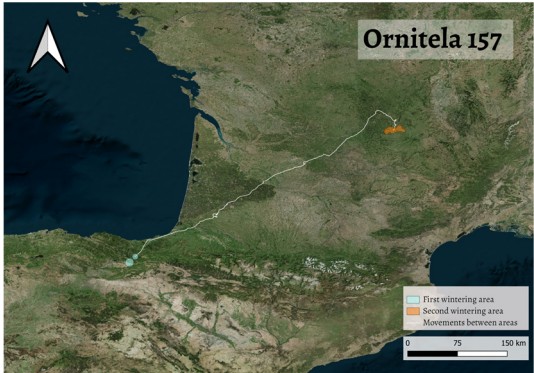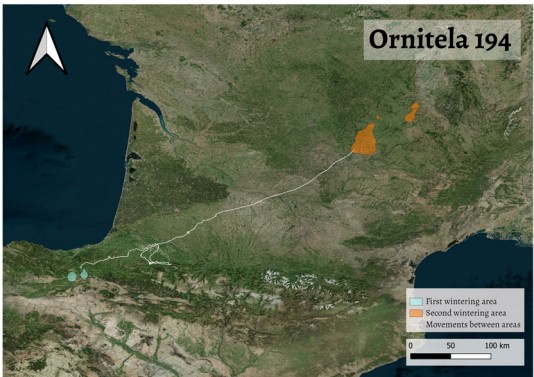

**Figure 2.** Area shifts performed by individuals during their non-breeding seasons. First (blue) and second (orange) wintering areas are represented for each season. The areas are drawn together in the map but could represent different years.

**Table 2.** Details of movements of the individuals with two wintering areas. All French localities were placed in the south of the region.

| Individual (Age) | Season | First Area | Days in First Area | Second Area | Days in Second Area | Travelling Days Between Areas | Distance Between Areas (km) | Direction | When Area Changed |
|---|---|---|---|---|---|---|---|---|---|
| Segovia 03 (A) | 2012/2013 | Segovia | - | Pamplona | 61 | 1 | 290 | S -> N | before spring migration |
| Álava 04 (A) | 2014/2015 | Southern France | 48 | Álava | 60 | 1 | 530 | N -> S | after autumn migration |
| Álava 05 (A) | 2016/2017 | Southern France | 45 | Southern France | 37 | 2 | 267 | N -> S | after autumn migration |
| Toledo 01 (A) | 2016/2017 | Toledo | 27 | Badajoz | 111 | 1 | 108 | NE -> SW | before spring migration |
| Ecotone 31 (I) | 2018/2019 | Huesca | >99 | Southern France | 23 | 1 | 187 | S -> N | before spring migration |
| Ecotone 106 (I) | 2018/2019 | Segovia | >66 | Salamanca | 23 | 2 | 207 | N-> S | before spring migration |
| Ornitela 79 (A) | 2019/2020 | Segovia | 25 | Pamplona | 28 | 5 | 250 | S -> N | before spring migration |
| Ornitela 154 (I) | 2019/2020 | Álava | - | Southern France | - | - | ~400 | S -> N -> S | wandering movements; first, it moved to the French area; then, it returned to first area before migration |
| Ornitela 157 (I) | 2019/2020 | Álava | - | Southern France | 43 | 2 | 465 | S -> N | before spring migration |
| Ornitela 194 (A) | 2019/2020 | Álava | - | Southern France | 27 | 10 | 412 | S -> N | before spring migration |

### 3.2. Home Range Size

Home ranges were highly variable in size (Table 1, Figure S1). MCP was estimated as $2173 \pm 3519$ km$^2$ (range = 27–20340); 95% KDE was estimated as $1158 \pm 2182$ km$^2$ (11–13176); 75% KDE as $438 \pm 899$ km$^2$ (5–4904); and 50% KDE estimated as $158 \pm 344$ km$^2$ (0.1 –2056). In Table 1 every parameter was broken down by sex, age and number of wintering areas, including mean, standard deviation, minimum and maximum values. The

largest area was 1000-fold larger than the smallest area (95% KDE of 11 km$^2$ vs. 13,176 km$^2$), with a great variability between (Figure 3).

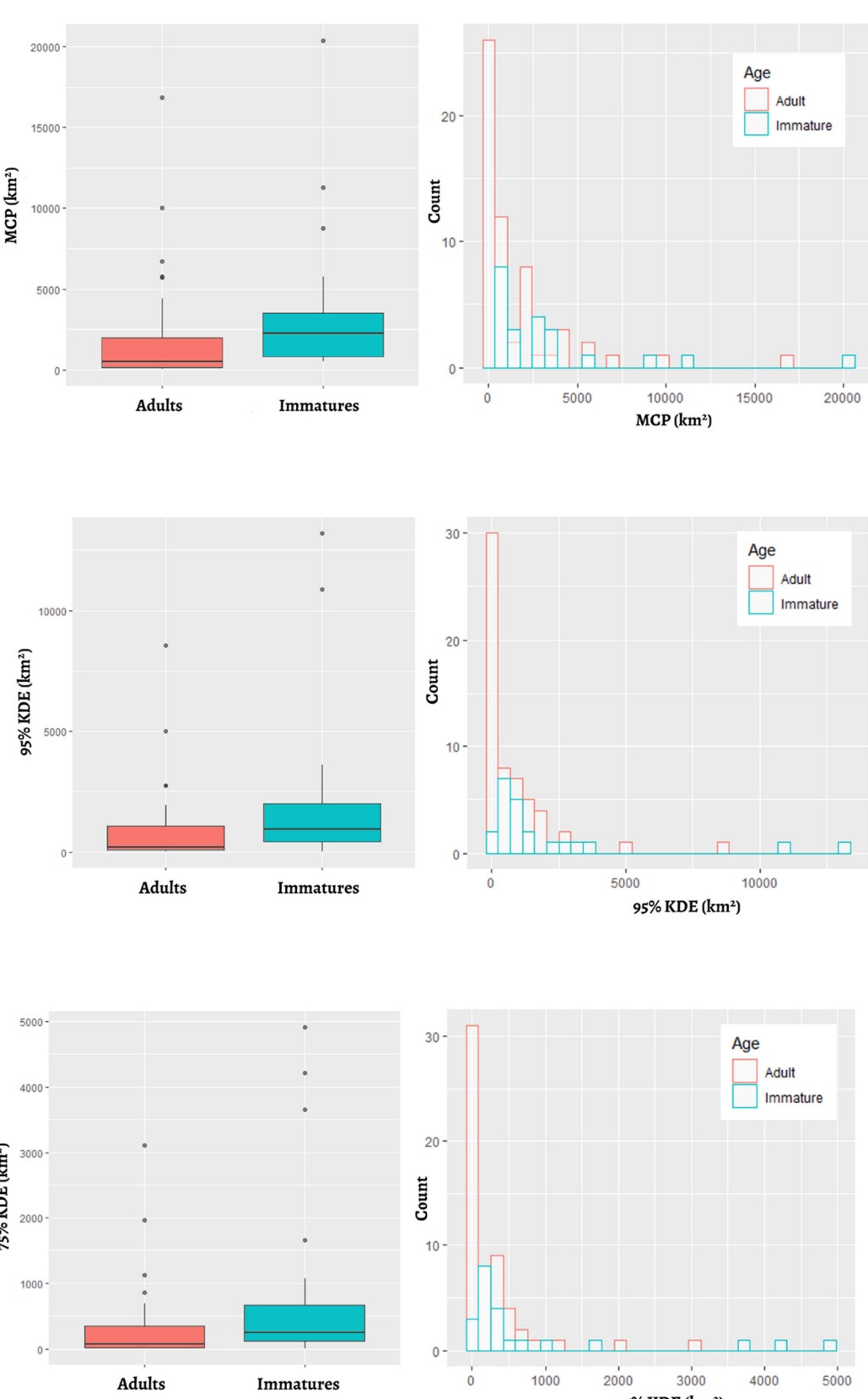

**Figure 3.** *Cont.*

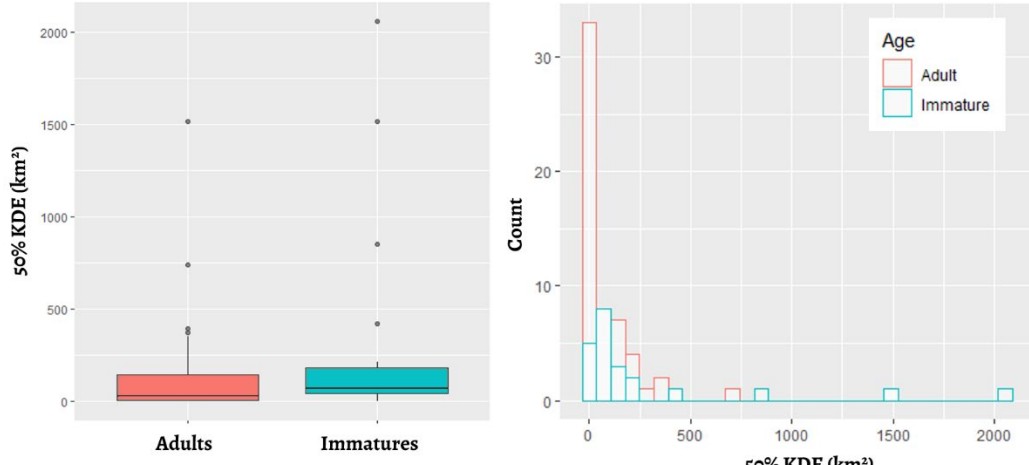

**Figure 3.** Comparison between age (immatures/adults) based on home range size estimations (MCP, 95% KDE, 75% KDE and 50% KDE). Boxplots are shown on the left (horizontal line—median; vertical line—minimum and maximum values without outliers; points—outliers). Histograms are shown on the right.

No effects of sex or number of wintering areas were found in the home range sizes (Table 3, Figures S2 and S3). However, we found significative differences between immatures and adults considering every home range estimation (MCP 95% KDE, 75% KDE and 50% KDE; Figure 3). All these parameters were at least two-fold smaller in immature individuals (Table 1, Figure 3, Figure S4).

**Table 3.** Results of the linear mixed models (LMMs) of home range areas (km$^2$). We used wintering area as the sample unit, so two areas during the same wintering season were considered as separated areas. Number of wintering areas; age; and sex were considered fixed factors. Wintering event and individual identity were considered random effects. Estimate, standard error (SE), degrees of freedom (Df), *t* value and *p* value are shown for each LMM. MCP—minimum convex polygon. α = 0.05. **Bold** indicates *p* Value < α.

| Fixed Factor (n) | Variable | Factor | Estimate | SE | *Df* | *t* Value | *p* Value |
|---|---|---|---|---|---|---|---|
| Number of areas (one/two) (*n* = 82). | **kernel 95%** | intercept no areas (1) | 1396.10 −400.88 | 303.86 296.08 | 32.33 61.27 | 4.595 −1.354 | **<0.001** 0.181 |
| | **kernel 75%** | intercept no areas (1) | 529.17 −140.40 | 128.25 123.46 | 24.46 55.65 | 4.126 −1.137 | **<0.001** 0.260 |
| | **kernel 50%** | intercept no areas (1) | 195.69 −63.10 | 47.84 46.64 | 30.26 59.76 | 4.090 −1.353 | **<0.001** 0.181 |
| | **MCP** | intercept no areas (1) | 2101.89 250.05 | 532.26 488.71 | 45.07 72.57 | 3.95 0.512 | **<0.001** 0.61 |
| Age (*n* = 82) | **kernel 95%** | intercept age (immatures) | 796.10 1315.10 | 277.50 529.20 | 78.0 78.0 | 2.869 2.485 | **0.0053** **0.0151** |
| | **kernel 75%** | intercept age (immatures) | 271.10 605.80 | 113.2 215.9 | 78.0 78.0 | 2.394 2.806 | **0.0191** **0.0063** |
| | **kernel 50%** | intercept age (immatures) | 111.42 170.93 | 44.29 84.45 | 78.0 78.0 | 2.516 2.024 | **0.0139** **0.0464** |
| | **MCP** | intercept age (immatures) | 1553.90 2140.75 | 528.37 861.46 | 44.93 77.61 | 2.941 2.485 | **0.005** **0.015** |

**Table 3.** *Cont.*

| Fixed Factor (n) | Variable | Factor | Estimate | SE | *Df* | *t* Value | *p* Value |
|---|---|---|---|---|---|---|---|
| Sex (*n* = 32) | **kernel 95%** | intercept | 889.215 | 333.246 | 10.521 | 2.668 | **0.0226** |
| | | sex (female) | −329.996 | 552.939 | 7.105 | −0.597 | 0.5692 |
| | **kernel 75%** | intercept | 283.624 | 121.982 | 10.886 | 2.325 | **0.0404** |
| | | sex (female) | −92.647 | 202.523 | 7.418 | −0.457 | 0.6604 |
| | **kernel 50%** | intercept | 95.548 | 45.931 | 11.033 | 2.080 | 0.0616 |
| | | sex (female) | −21.549 | 76.115 | 7.463 | −0.283 | 0.7848 |
| | **MCP** | intercept | 2664.53 | 1144.579 | 11.143 | 2.328 | **0.0397** |
| | | sex (female) | −1982.23 | 1982.92 | 9.382 | −0.987 | 0.3484 |

## 4. Discussion

In this study, we analysed the wintering season of red kites in Spain using a large sampling effort and satellite/GPS telemetry, contributing to a better knowledge of the spatial strategies of red kites during this important period of their annual cycle. Our work demonstrated a dual strategy of the red kites' wintering in Spain: most individuals remained in only one area, but some of them changed their wintering area within one wintering season, without increasing the duration of the wintering season. Furthermore, we found a high variability in the home range sizes of the wintering red kites. There were no differences according to sex or number of areas used, but adults had smaller home ranges than immatures. Our results highlight the intraspecific variability and plasticity of the red kite during the wintering season.

The beginning and end dates of the wintering seasons of our individuals, as well as their durations, were within the expected range, considering previous works on red kites [36]. The red kites also had a similar wintering durations to other raptors that spend their winters in Africa [9,37]. The long-term study of the phenological behaviour of raptors is of great interest in understanding the capability of these species to respond to climatic changes [38]. Future studies about the phenology of the red kite could be focused on how their distribution patterns and wintering parameters are influenced by global warming, especially considering that Spain, its main wintering destination, is at the southern limit of its distribution. These circumstances may mean that the Spanish wintering population will shift their wintering quarters farther north in the future, staying within the envelope where winter conditions are suitable. We recorded wintering quarters shifts performed by some of the red kites. Although most individuals spent the whole winter in the same area, 23% of them changed the area in the middle of the season, travelling hundreds of kilometres to a new location. Furthermore, this behaviour had no continuity over time because individuals performed area shifts just once, and the rest of their recorded years were spent wintering within only one of those areas. Similar displacement during winter was reported by Pfeiffer and Meyburg [36]; a change in winter quarters of over 130 km was performed by an adult female at the end of December. Furthermore, this behaviour has also been demonstrated in other raptors species, such as the booted eagle *Aquila pennata* [9] or the osprey *Pandion haliaetus* [39]. Our results demonstrated that this behaviour is commonly present in the red kite, even when they still have high fidelity for their wintering quarters. More specific studies could investigate the ecological factors that drive these area shifts in the red kite and other raptors. This strategy may be driven by food or habitat resource availability, in association with climate change [10]. Age and sex were found to be unimportant in these area shifts. Age was, however, an important determinant of range size, with adults occupying a significantly smaller range than juveniles. Our results also showed that adults' home range sizes are two- or three-fold smaller than immatures'. Furthermore, even adults decreased their home range areas until they were stabilized, usually after the second or third year of life. The reduction in home range sizes, common among raptors [40], could indicate a better knowledge of the territory and ability to hunt, which entails greater efficiency in energy expenditure

The home range sizes varied greatly between individuals. This variability was also found in the booted eagle in Africa [9], which showed 95% kernels of 398 km$^2$, 75% kernels of 132 km$^2$ and 50% kernels of 72 km$^2$, considerably smaller than in the case of the red kites observed in the current study. On the other hand, *Aquila pomarina*, a species that also over-winters in Africa, showed larger overall wintering home ranges (up to 112,000 km$^2$) [41]. This intraspecific variability may be explained by the possibility of expanding or restrict-ing the range area depending on the availability of food and resources, as well as by the geographical and biological characteristics of the territory selected by the individual. Furthermore, the changes could also be due to the physiological state of the individual, especially when they are infected by parasites or affected by pollutants [21,42]. Interspecific differences, on the other hand, may also be due to the particularities of the territory selected by the species during wintering and, above all, its biological capabilities and ecological requirements (amount of food required, fat storage capacity, feeding efficiency, etc.) [2].

**Supplementary Materials:** The following are available online at https://www.mdpi.com/article/10.3390/d14020117/s1, Figure S1: Wintering areas (95% KDE) used by the 44 tagged Red Kites during their non-breeding seasons in Spain and Southern France. Each color represents one wintering area, Figure S2: Comparison between individuals with one and two areas during the same wintering event based on home range size estimations (MCP, 95% KDE, 75% KDE and 50% KDE). Figure S3: Comparison between sexes based on home range size estimations (MCP, 95% KDE, 75% KDE and 50% KDE). Figure S4: Wintering areas (95% KDE) used by immatures (green) and adults (red) Red Kites during their non-breeding seasons in Spain and Southern France. Table S1: Parameters of 44 Red Kites tracked by GPS/satellite telemetry tagged in Spain during the wintering season. Sex (F: female, M: male) and Age (A: adult, I: immature) are indicated together with bird ID. The wintering province in Spain and the summer destination after the spring migration are also shown. A hyphen (-) in-dicates that information is not available. Italics on dates indicate that they correspond to the be-ginning/end of measuring, not to beginning/end of wintering, so those were considered as "partial" winterings.

**Author Contributions:** Conceptualization, J.G.-M., J.D.L.P., A.B.-B. and V.U.; methodology, J.G.-M., J.D.L.P., A.B.-B. and R.R.; formal analysis, J.G.-M.; investigation, J.G.-M.; writing—original draft preparation, J.G.-M.; writing—review and editing, J.D.L.P., A.B.-B. and V.U.; supervision, V.U.; Project administration: J.D.L.P., A.B.-B. and R.R. All authors have read and agreed to the published version of the manuscript.

**Funding:** This research received no external funding.

**Institutional Review Board Statement:** Not applicable.

**Data Availability Statement:** Original GPS locations are available at movebank.org under authors permission.

**Acknowledgments:** Information about tagged red kites was obtained within the 'Migra' program developed by SEO/BirdLife, with the collaboration of Fundación Iberdrola España. We are very grateful to all volunteers that helped during all the fieldwork. Specially, we want to thank for their help to: Ángel Arredondo, Ángel Gómez, Angélica Muñoz, Arturo Rodríguez, Carlos Ponce, David Camacho, Emilio Escudero, Gorka Belamendia, Iñaki Martínez, Javier Frías, Javier León, José Polanco, Joseba Carreras, Joseba Markinez, Joseba Sánchez, Juan Antonio García, Juan Carlos Perlado, Lorena Alos, Manuel Aguilera, Marta Olalde, and Virginia de la Torre. The tagging of the birds was developed thanks to the collaboration and/or funding of: Fundación Iberdrola España, TB Raab, Hontza, Diputación Foral de Araba, Gobierno del País Vasco, Fundación Biodiversidad, SEO-Monticola, Fondo de Amigos del Buitre, and Ayuntamiento de Binaced. We are also thankful to Diputación Foral de Araba, Gobierno de Aragón, Junta de Castilla y León, and Junta de Castilla-La Mancha. This paper is part of Jorge García-Macía's phD.

**Conflicts of Interest:** The authors declare no conflict of interest.

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
