# Peer review of "High Variability and Dual Strategy in the Wintering Red Kites (Milvus milvus)"

_diversity, doi:10.3390/d14020117_

Round 1
Reviewer 1 Report
This is a useful, well conducted and appropriately analysed study of a little known aspect of Red Kite biology - and one that is not easily tackled. I detected no major issues. The paper's main flaws are related to English expression and I have suggesd in detail how this might be addressed.
Title: As far as I can tell “Estrategy” is a business term, “Strategy” would be appropriate here. I also suggest condensing the title to:
Wintering Strategies of the Red Kite (Milvus milvus)
General
Capitalise “red kite “ throughout (including references).
Abstract
Line 9 delete this sentence – the paper is not really about population dynamics
Line 10 replace ‘to study’ with ‘to develop effective conservation strategies’ and replace ‘therefore mandatory’ with ‘essential’
Line 12 delete ‘opportunistic’ - unless something is added about in what way the kite is opportunistic
Line 16 delete the first comma and change ‘no’ to ‘not’
Line 18 change ‘areas’ to ‘area’
Line 20-21 I suggest: “adults had home ranges two to three times smaller than those of immatures” OR ‘”The homes ranges of adults were two to three times smaller than those of immatures”
Introduction
Line 27 replace ‘in’ with ‘on’ and ‘performed by’ with ‘of’
Line 29 “replace with ‘Satellite/GPS telemetry has increased understanding of how individuals’
Line 30 replace ‘in’ with ‘on’
Line 32 replace ‘or’ with ‘and’
Line 36 replace ‘opportunistic’ with ‘partially migratory’ or similar term
Line 40 replace ‘it’ with ‘the species’
Methods
Lines 54-57 replace ‘Some countries are increasing their breeding and wintering populations (with a progressive sedentarization), but some other 55 countries are decreasing them, specially those placed in its southern distribution limits, which currently held the main wintering populations, like Spain [24]. The global warming’ with
‘In some countries, breeding and wintering populations are increasing and becoming progressively more sedentary, but elsewhere, especially at the species’ southern distribution limits, including Spain, which currently supports the main wintering populations, numbers are decreasing. Global warming…’
Lines 59-63 replace ‘For this reason, the study of the variability in the wintering strategies of the red kites is mandatory to understand the behaviour of the species, as well as knowing its conservations prospects and elaborating accurate management plans.
Here, we tagged 44 Red Kites” with
.For this reason, the study of the variability in the wintering strategies of the Red Kites is mandatory to understand the behaviour of the species, and its conservation needs..
We tagged 44 Red Kites’
Line 66 replace ‘end dates, and locate their wintering areas; (2) to characterize these wintering areas using’
With
‘end dates (2) to locate and characterize wintering areas using’
Line 69 replace ‘possible different strategies between age and sexes.’ With
‘possible different strategies according to age and sex.’
Line 76 replace’ and 31 individuals with not determined sex with
‘and 31 individuals whose sex was not determined’
Line 77 replace ‘season, and adults after the second’ with
‘season, and adults from the second’
Results
Line 129 replace ‘were not available’ with ‘was not available’
Lines 136-7 replace ‘They spent on average 96 27 days 136 in their wintering areas, in a range between 32 and 139 days (Table 1)’ with
‘They spent between 32 and 139 days in their wintering areas, averaging 96 ± 27 days (Table 1).
Line 145 replace ‘individuals returned to the same wintering area after the autumn migration in the 89% of total seasons’ with
‘returned to the same wintering area in 89% of seasons’
Line 150 replace ‘no’ with ‘not’
Line 157 change ‘Furthermore, it was performed by both adults (6) and immatures (4).’ to ‘Both adults (6) and immatures (4) used two wintering areas’.and move to line 154 after sentence ending France (n=1).
Line 172-5 change ‘The ten individuals that changed areas during winter did so mainly at the beginning of winter, after autumn migration; and at the end of winter, prior to spring migration, when they returned 174
to their nests or summering locations in Central Europe.
Line 179 ‘3.2 Home range size’
Lines 180-3 change to
‘Home ranges were highly variable in size (Table 1): MCP was estimated as 2173 3519 km2 (range= 27– 20340 ); 95% KDE as 1158 2182 km2 (11 - 13176), 181;75%; and KDE as 438 899 km2 (5 - 4904) km2 (0.1 - 2056).’
Line 1865replace ‘bigger’ with ‘larger’
Line 186 remove ‘them’
Line 199 insert ‘(in km2)’ after ‘home range areas’.
Table 3 lower case ‘df’
Was there interaction between age and sex or were too few sexed to determine that?
Discussion
Line 211 replace ‘lifecycle’ with ‘annual cycle’
Lines 211-3 delete from ‘Previously’ to ‘14’;, repeats introduction
Lines 216-8 change to ‘variability in home range sizes of the wintering Red Kites. There were no differences according to sex or number of areas used, but adults had smaller home ranges than immatures.’
Line 222-3 change to ‘The kites also had a similar wintering duration to other raptors that spend their winters in Africa [9, 32]’.
Lines 227-30 replace ‘its main wintering destination, is placed in the southern limit of its distribution. These circumstances may do that the Spanish wintering population place their wintering quarters in northern areas in the future, where they have better environmental conditions during the winter.
We reported wintering quarters shifts performed by some of’ with’
‘its main wintering destination is at the southern limit of its distribution. This circumstance may mean that the Spanish wintering population will shift their wintering quarters farther north in the future, staying within the envelop where winter conditions are suitable .
We recorded wintering quarters shifts performed by some of’ with’
Lines 236-9 Change ‘’ These displacements during winter were reported by Pfeiffer & Meyburg [31]: a change in winter quarters of over 130 km was performed by an adult female at the end of Decem- ber. Furthermore, this behaviour was also demonstrated in other raptors species like the Booted Eagle Aquila pennata [9] or the Osprey Pandion haliaetus [34].’ to’
‘Similar displacement during winter was reported by Pfeiffer & Meyburg [31]: a change in winter quarters of over 130 km was performed by an adult female at the end of December. Furthermore, this behaviour has also been demonstrated in other raptors species such as the Booted Eagle Aquila pennata [9] and Osprey Pandion haliaetus [34].
Line 242 close gap between ‘shifts’ and ‘in’
Lines 241-4 chsnge ‘More specific studies could approach the ecological factors that drive these area shifts in the Red Kite and other raptors. This strategy may be driven by food or habitat resources availability [10], due to either influences of age, sex or other important condition or life history factors were not found in our study.’ To
‘More specific studies could investigate the ecological factors that drive these area shifts in the Red Kite and other raptors. This strategy may be driven by food or habitat resources availability, in association with climate change [10]. Age and sex were found to be unimportant in our study.
Lines 243-4 delete ‘due to either influences of age, sex or other important condition or life history factors were not found in our study.’
Lines 245. Insert ‘Age was, however, important determinant of range size, with adults occupying a significantly smaller range than juveniles.’ Follow by lines 259-263.
Then address high variability (lines 245-258). Reference needed for last sentence (lines 254-8); Newton 2008 should cover it.
Line 250 remove “This’.
Line 256 replace ‘its’ with ‘the species’
Reviewer 2 Report
In this manuscript, the authors describe wintering strategies of the Red Kite. Field work was conducted in Spain. Number of studied birds is impressive – 24 adults and 20 immatures. Good job! The article is interesting, well written and comprehensive. However Methods and Results should be described in more details and clearly. My specific comments follow.
Detailed comments:
Title: Estrategy or strategy? What does Estrategy mean?
Introduction/Discussion: Please give some general information about Red Kite, it will be interesting for non-specialists. Give some information about habitat in which Red Kites are wintering in Spain and something about food and foraging. Are there any special feeding programs in wintering areas?. Does Red Kite exhibit Sexual Size Dimorphism? How differences in size between sexes can influence lack differences in wintering strategy between sexes?. You can cite: David, T. S., J. Orta, D. A. Christie, and E. F. J. Garcia (2021). Red Kite (Milvus milvus), version 1.1. In Birds of the World (S. M. Billerman, Editor). Cornell Lab of Ornithology, Ithaca, NY, USA. https://doi.org/10.2173/bow.redkit1.01.1 or literature included there.
Line 73-74: move to results section.
Line 76: give the age (ad., imm.) of sexed birds.
Line 84: 5% is not recommended now, see page 952 in Bodey, T. W., Cleasby, I. R., Bell, F., Parr, N., Schultz, A., Votier, S. C., & Bearhop, S. (2018). A phylogenetically controlled meta‐analysis of biologging device effects on birds: Deleterious effects and a call for more standardized reporting of study data. Methods in Ecology and Evolution, 9(4), 946-955. Please add in line 84 „5% [29] but see [Bodey et al. 2018 – give exact number of paper].
Line 85-90: Did you check the influence of different intervals (number of GPS points for each individuals) on home range estimations? , i.e. do home range estimations differ between tags which collected locations every hour and these which collected data points every 5 minutes?
Line 99-100: why the date of bird returned to Central Europe was consider as the end of wintering period?. It is odd. It seems more reasonable that the end of wintering period should be the last day of staying on wintering ground. If I understand correctly Authors include spring migration in wintering period? Please rewrite if I did not understand it correctly – be more specific.
Line 116: Please, give number of levels in each factor, eg. “Number of wintering areas (1, 2)”, sex (F, M). It is obvious but be more specific.
Line 117: What does exactly “wintering event” mean
2.2. Spatial parameters and analysis – please describe what was the exact sample size in each analysis. Sometimes it is hard to follow it in Results, describe it in Methods.
Figure 1. Is it possible to color an area for the same individual with the same color? Maybe split the maps for immatures and adults and color the area? This map (Figure 1), in current version shows areas where birds were wintering. It is ok, but more interesting will be to show individual areas. Partially it was done in Figure 2 – nice picture. It will be also interesting to see wintering areas of birds which did not change wintering places, to see adults and immatures home ranges. Give more interesting maps, it could be in Supplementary Materials.
Figure 2. Is blue for first wintering area and orange for second? Please describe. Each category should be in different color.
Line 135: 17 November – how it was calculated, describe in Methods
Line 136: end of January and the first week of June – why so huge differences? Is it for adults and immatures together? Give date for adults and immatures, cee comment below.
Line 134-137: Some comparison of date of arrivals and date of departure between sexes, years, age of birds will be interesting. Why the Authors did not do it? Whether the Authors consider these analyzes?
Line 145: give the sample size, describe in Methods how it was calculated.
Line 148-149: How it was exactly calculated, describe in Methods. Did you include only wintering event with full bird presence, i.e. did you exclude wintering event when bird was trapped? – bird could change wintering place before it was trapped – be more specific. Explain it also in the context of results included in Table 3 and in the context of statistical analysis – line 110-116. What does exactly sample size = 82 and sample size = 34 mean? Full wintering events?
Line 157-160: hard to follow, please rephrase and write it more clearly.
Figure 3. Axis descriptions are too small, figures are not legible. If Adults are blue on boxplots they should be also blue on histograms. Categories overlap on histograms which is not legible. What does points and line on boxplot exactly mean? – be more specific.Line 199: A) – rather a, this description is not clear.Supplementary materials: what does Beginning and End mean, beginning of wintering? End of wintering? End of spring migration? Be more specific.
Line 199: “significant differences between immatures and adults in home ranges” – please discuss it in Discussion section – was this pattern found in other birds of prey?
